# Operational Health Pavilions in Mass Disasters: Lessons Learned from the 2023 Earthquake in Turkey and Syria

**DOI:** 10.3390/healthcare11142052

**Published:** 2023-07-17

**Authors:** Roberto Scendoni, Mariano Cingolani, Vittoradolfo Tambone, Francesco De Micco

**Affiliations:** 1Department of Law, University of Macerata, 62100 Macerata, Italy; r.scendoni@unimc.it (R.S.); mariano.cingolani@unimc.it (M.C.); 2Research Unit of Bioethics and Humanities, Department of Medicine and Surgery, Università Campus Bio-Medico di Roma, 00128 Roma, Italy; v.tambone@unicampus.it; 3Department of Clinical Affair, Fondazione Policlinico Universitario Campus Bio-Medico, 00128 Roma, Italy

**Keywords:** mass disasters, earthquake, disaster management, safety, emergency, health security, survivors, public health

## Abstract

The massive earthquake that hit Turkey and Syria in February 2023 killed tens of thousands of people, and most of the deceased have not yet been identified. Many victims were pulled from the rubble hours or days later, injured and in need of assistance, treatment, and food, and many have not yet been connected with their families. Armed forces, volunteers, technicians, and health workers must cooperate in synergy in these situations to ensure effective interventions and to improve resilience. Based on the lessons learned from the response efforts to this recent natural catastrophe, this brief report proposes, for the first time, an organisational model structured around five functional pavilions that can be safely set up at the edge of a disaster area. Each pavilion should run its own activities to make a vital contribution to the overall coordinated emergency response. Looking to the future, it is extremely important to apply a technical approach that leads to maximum operational synergy at a disaster site and during the first phase of a sudden-onset emergency.

## 1. Introduction

On 6 February 2023, a powerful earthquake (more precisely two big earthquake tremors) hit southern Turkey and northern Syria, followed by hundreds of aftershocks.

The earthquakes destroyed thousands of homes in Turkey and Syria, while hospitals, roads, and bridges were badly damaged. A 7.8 magnitude earthquake is completely devastating, and all the healthcare workers in the area—and often an entire population—will be involved in the disaster response. The first earthquake, with a hypocentre at a depth of approximately 17.9 km and an epicentre 34 km northwest of the city of Gaziantep, registered a magnitude of 7.8 Mww. Subsequently, a new tremor, with a magnitude of 7.5 Mww and an epicentre located 4 km south of the city of Ekinözü, mainly affected the territory of the province of Kahramanmaraş. However, large hospitals may not always be close to a disaster site and resources may be scarce. Therefore, emergency medical services (EMS) play a vital role in the immediate response to a mass disaster.

More than 46,000 people have died in Turkey, while more than 7000 people are estimated to have died in Syria. After more than three months, nearly 2.7 million people are still living in tents; to date, the wounded are more than 100,000. Thousands of survivors have been put at risk due to the destruction of infrastructure and freezing temperatures in the affected areas from the very beginning. Efforts immediately after the earthquakes and in the days following focused on search and rescue, to locate survivors in the rubble of collapsed buildings [1].

Health systems, even interim ones, are at the heart of disaster resilience because it is critical to deliver health services to devastated communities in a timely manner [2]. It is also essential to ensure accurate body identification and support survivors and the families of victims by any means. 

Due to uncertainty with regard to the extent of the damage, disruption, and a lack of reliable information, there is often difficulty in establishing an immediate emergency response following a disaster. However, coordination at all levels (local, regional, national, and/or international) is imperative. Although disaster response plans often provide for corresponding coordination mechanisms, these may not exist immediately following a disaster [3]. In order to save the lives of as many wounded as possible, the limited medical resources on the ground must be utilised rationally by classifying and processing the mass of wounded and determining which life-threatening injuries must be treated as a priority [4]. In addition, a coordinated international effort is needed to speed up the recovery and identification of victims. In this context, reference intervention models must always be adapted to the specific situation, and the personnel involved must always cooperate in synergy.

## 2. Methods

Managing a mass disaster, in particular, an earthquake [5], requires a holistic approach involving multiple factors and resources. Here is a general method that can be followed to manage a mass disaster.

### 2.1. Preventive Planning

Identify and assess the potential risks and disasters that may occur in your area. Develop a detailed emergency plan that covers a wide range of disaster scenarios. Engage the relevant stakeholders, such as government agencies, law enforcement, medical services, and volunteer organisations, in the planning process.

### 2.2. Preparation

Organise regular drills to test the effectiveness of the emergency plan and train personnel involved in disaster response. Establish a reliable communication system to rapidly and effectively inform the public and the involved staff. Acquire and maintain the necessary resources for disaster management, such as medical equipment, food supplies, clean water, power tents, fuel, etc.

### 2.3. Immediate Response

Activate the emergency plan as soon as the disaster occurs. Implement evacuation measures, if necessary, to ensure the safety of the affected individuals. Provide immediate medical care to victims and ensure medical facilities are established to treat the injured.

### 2.4. Coordination and Resource Management

Establish an emergency operations centre to coordinate response activities, ensuring smooth communication among all the involved parties. Assign specific roles and responsibilities to different teams and agencies to avoid overlaps or a lack of action. Efficiently manage the available resources by allocating them where they are most needed and keeping track of all the supplies.

### 2.5. Long-Term Assistance and Recovery

After addressing the immediate response phases, provide long-term assistance to the affected individuals, such as temporary housing, counselling services, and psychological support. Furthermore, reinstate basic services, such as water, power, and transportation.

The management of the scenario may vary depending on the type of event, the resources available, and the specific context. Therefore, it is crucial to adapt the method based on the needs and circumstances of each emergency situation [6].

Our proposal arises both from what is reported in the scientific literature and from one’s own professional experience. In particular, the authors participated in inspections and post-mortem investigations on the occasion of mass disasters that occurred in Central Italy. We also took part in national and international conferences on the topic.

Taking into account the events that have occurred and the research that is still being conducted [7,8,9,10], the authors propose an organisational model structured around five functional pavilions, which can be safely set up at the borders of a disaster area. Based on the Disaster Mortuary Operational Response Teams, each of the five pavilions performs an operational function [11]. Each pavilion would conduct its own activity, to contribute in a vital way to the overall coordinated emergency response (Figure 1). 

## 3. Pavilion for Armed Forces Personnel and Excavation Technicians

This is the first pavilion that should be set up. The military is traditionally called upon to assist in the search and rescue phase of a disaster management cycle. In this stage, the timely, effective, and efficient deployment of Operational Response Teams can pay off in terms of damage limitation and saving lives. Expert personnel must be involved in the recovery of victims of mass disasters, as well as technicians who make available the necessary tools and means of carrying out evacuation work and recovering victims from the rubble [12]. In recent years, there has been a marked improvement in coordination between military units/formations and local civil administration.

Readiness, adequate training of the armed forces, and the resources—financial and otherwise—possessed by the armed forces are required. It is also mandatory to coordinate efforts if the military representation is multinational.

However, much remains to be accomplished to fill in the gaps and achieve the desired synergy between local civilian and military resources to optimise the outcome of disaster relief efforts. The undocumented removal of remains by volunteers, continuous and unsupervised access to the site and autopsy area, the recovery of remains by family members prior to identification, non-involvement of forensic personnel, and logistical criticalities occurred at the disaster site [13]. The looting phenomena that often occur during a response phase must also be prevented.

Body recovery is often performed spontaneously by a large number of individuals, so coordination of all the involved groups is needed to encourage the use of the recommended health and safety precautions. Bodies should be placed in body bags. If these are unavailable, use plastic sheets, shrouds, bed sheets, or other locally available material, and body parts (e.g., limbs) should be treated as individual bodies. Recovery teams should not attempt to match the body parts at the disaster scene. Personal belongings, jewellery, and documents should be separated from the remains during the identification phase, not before, and ambulance services should not be employed for this purpose as they are already busy helping the living during the recovery process. Finally, temporary burial sites should be constructed to help ensure the future location and recovery of bodies [14].

## 4. Health Care Pavilion

Given the magnitude of the earthquake, all healthcare workers and the entire population of the area are involved. As EMSs respond to all types of man-made and natural emergencies, hazards, and disasters, healthcare personnel often work hand-in-hand with public safety specialists and law enforcement agencies. The EMS should assist in the transport of individuals to safe places where first aid can be provided, especially for people with injuries or fractures who need to be mobilised by experienced professionals [15]. A pavilion needs to be set up quickly near the affected area, with medical, nursing, and social health personnel capable of providing primary care. The appointment of a health supervisor is essential for coordinating the triage sectors, regulating the flow of injured people, and managing the allocation of resources [16]. Many patients will be transported to major healthcare facilities, which should be provided with preliminary information on the status of arriving patients, with initial triage already being conducted at the disaster site. In this way, resource wastage, overcrowding, and delays in treatment can be avoided. An ideal crisis response model is an integrated service that involves EMS and law enforcement to provide support to people experiencing a health crisis, such as one that occurs after a mass disaster.

## 5. Post-Mortem Pavilion

One of the most difficult components of disaster response is the proper management of the dead. It is a key pillar of emergency response, with long-lasting and significant consequences for the survivors and communities. Disasters kill tens of thousands around the world each year. Even so, the care of the deceased is often neglected in disaster planning. Major disasters in recent years, such as the 2010 Haiti earthquake or the 2016 Central Italy earthquakes, have shown that there is not enough guidance for first responders [17]. Immediately following a major disaster, residents from local communities are the first to deal with the retrieval of dead bodies, before forensic specialists are able to reach the site. The nature of a catastrophic event makes the correct recovery of human remains extremely difficult as well as the search for ante-mortem information necessary for the identification of dead bodies. The early work of non-specialists in managing the dead (especially proper recovery, documentation, and storage methods) will determine much of the success of future identifications by forensic specialists. It is important to set up a team to carry out external examinations and autopsies, if necessary, along with the collection of information useful for identification purposes (e.g., body marks, skeletal or dental morphological characteristics, and fingerprints) [18,19]. This team often liaises with another team whose responsibility is to contact possible relatives and acquaintances of victims. This effort enables the collection of all the identity data. It is useful to collect and catalogue biological samples that can be used for DNA analysis. Forensic pathologists and forensic anthropologists should cooperate in these activities through standard methods and innovative approaches [20]. Digital cameras make it easier to store and distribute photographs. The body must be adequately cleaned so that photographs are able to properly represent facial features and clothing. It is then advisable to use mobile radiographic equipment capable of performing radiological analyses for the dual purpose of defining the cause of death and identifying the victim [21]. In addition, X-ray examinations provide valuable information pertaining to different population groups and for the estimation of age [22]. Finally, special personal identification forms should be filled in, such as those issued by Interpol DVI (Disaster Victim Identification) [3]. A dead body should only be released when the identification is certain. The following final recommendations are suggested for this pavilion:-visual recognition should be confirmed by other information, such as the identification of clothing or personal effects;-information collected about missing people can be used to cross-check visual recognition;-a body should only be released by the responsible authority, which must also provide documentation of the release (a letter or death certificate).

From a medicolegal perspective, there are some main objectives with regard to the study of human remains: a good scene of crime investigation, with the proper retrieval and registration of human remains, and the application of proper identification procedures; adequate autopsy techniques along with tissue sampling (e.g., toxicology) to eventually reconstruct the cause and determine the manner of death. It may be very difficult, depending on the country and authorities involved, to place these activities in a set protocol. For example, in countries in which DVI teams are well organised and Interpol protocols are known to all the specialists involved, the application of preset logistics and methodologies is fairly easy and automatic. In many other countries, this modus operandi may not exist [23].

Body recovery is often conducted spontaneously by a large number of individuals, so coordination of all the involved groups is needed to encourage the use of the procedures and health and safety precautions recommended in this manual. Bodies should be placed in body bags. If these are unavailable, use plastic sheets, shrouds, bed sheets, or other locally available material, and body parts (e.g., limbs) should be treated as individual bodies. Recovery teams should not attempt to match the body parts at the disaster scene. Personal belongings, jewellery, and documents should be separated from the remains during the identification phase, not before, and ambulance services should not be employed for this purpose as they are already busy helping the living during the recovery process. Finally, temporary burial sites should be constructed to help ensure the future location and recovery of bodies [24].

## 6. Pavilion for Food Distribution Volunteers

All governments consider the food supply chain (FSC) to be a critical infrastructure, and multiple strategies have been proposed to make the FSC more resilient to disruption [25]. However, major disasters, such as the earthquake in Turkey and Syria, have exposed vulnerabilities in the FSC. Preventing malnutrition in the population affected by a disaster is a primary objective, together with health care, especially in view of the consideration that many people are recovered several hours or days after the disastrous event. The issue should always be addressed at both a national and supranational level. However, it is necessary to plan ahead with regard to the treatment and management of cases of malnutrition that may have pre-existed the disaster, which may manifest during the relief operation. It will be necessary to (a) estimate the amount of food available; (b) calculate the food needs of the population concerned; and (c) determine food rations according to the characteristics of the population and the estimated duration of the effects of the disaster. In this activity, the participation of organised bodies, such as the Civil Protection Service, is important. When a tragedy of this size strikes, there is a good risk that victims may go without food and water for several days. Drones are quite helpful in these situations for the quick delivery of food packages and water bottles. Drones can assist in providing first assistance to those who require it, in addition to delivering food supplies [26]. 

## 7. Pavilion for Psychologists and Family-Finding Volunteers

Here, we are referring to immediate psychological support that is intended to help the survivors and relatives of victims to keep as calm as the situation permits and avoid panic. In this phase, the presence of competent and expert psychologists is essential; part of their mission is to provide first aid and reassure those who have been affected by a disaster and inform them about what is being achieved in the search for missing persons. Stress management in the initial stages can prevent scenarios of mass despair in the days that follow a catastrophe [27]. All the volunteers responsible for tracing the families of survivors (by telephone, video call, or email) and seeking immediate reunification among those who have been pulled alive from the rubble should also be placed within this area. Major associations, such as the International Red Cross and Red Crescent Movement, have been particularly active in this regard. Arranging funds to implement this activity will allow for better care management and better support for those affected. 

Local centres play an important role in the collection and consolidation of information on the dead. They assist the public in many ways, for example, by responding to tracing requests and releasing information about persons found or identified. In a disaster situation, a national system should be set up to centralise and manage all the information on the dead and missing persons. A system should be developed and coordinated by “formal” responders and pre-trained volunteers that can integrate a large number of spontaneous volunteers. This strategy essentially transforms spontaneous volunteers (individuals with or without specialised skills) into an assigned resource. We have seen this strategy implemented by individual organisations, but volunteer needs and resources are less often coordinated across organisations. There are often barriers to interorganisational coordination, such as differences in terminology, procedures, and operating structures. Potential volunteers may have to search several organisations or divisions before being able to contribute their skills. In this way, volunteers can respond to authoritative direction and act responsibly within their assigned area [28].

## 8. Discussion and Conclusions

As a vital component of the areas discussed, it is crucial to prepare the health sector to deal with all types of disasters. There is an ongoing need to create awareness of the associated public health risks, such as infectious diseases, and enhance the knowledge and skills of all the health actors involved in the response and recovery processes. The technical aspects of disaster preparedness include information dissemination and management; mass casualty management; damage and needs assessment; and humanitarian supply management. 

Operational synergy in a mass disaster is sometimes very difficult to achieve. In the absence of a common language and shared procedures, health teams, technicians, law enforcement agencies, and voluntary personnel risk finding themselves operating in difficult conditions, each pursuing its own objective or its own operational logic. The action that takes place in inaccessible places, the presence of any additional risks, the difficulties associated with accessing the victims, the climatic conditions, and the possibility of effectively channelling resources to the site of the event represent the binding aspects that must be considered in the management of the intervention. First aid and rescue teams must be properly trained. How can this integration be achieved between professionals? It is essential to place a coordinator or Disaster Manager (DI.MA.) on site who will have the task of setting up an advanced command point, optimising the available resources, guaranteeing communication links and supplies to the functional work areas, and verifying the safety of the operators. In this way, it will be possible to create a joint and cooperative action between the various pavilions, with the common aim of immediately guaranteeing initial post-disaster resilience.

After each disaster, a critical review of the activity should be carried out. This would provide an opportunity to learn from mistakes and improve forensic and medical work in mass disasters. In this perspective, the logistical-functional proposal formulated in this report is a proposal for improvement, to standardise the multidisciplinary activity that characterises these scenarios.

An international task force is needed to adopt new resilience measures that take into consideration the lessons learned from the consequences of the earthquake in Turkey and Syria. To alleviate the tension of being faced with sometimes unprecedented challenges, international and multidisciplinary collaboration is necessary, and a health audit will be required. Each country should draw up a list of professionals and volunteers of various types from the national network who can make themselves available to take to the field in emergency situations: a sort of “call to arms” with protection guarantees (insurance, job retention, etc.). Of course, the integrative commitment of voluntary associations and non-profit organisations on the front line is always a milestone. In responses to previous natural disasters, it has been shown that it is possible to adopt an integrated approach across regions and countries when it comes to law enforcement, but in the case of medical and healthcare professionals (physicians, pathologists, anthropologists, nurses, psychologists, etc.), it is more difficult to achieve integrated strategies. There is no standardised modus operandi in the event of a mass disaster, the characteristics of each disaster being completely unique; however, it is extremely important to apply a technical approach that leads to maximum operational synergy.

The requested task force should not be different from those already activated in previous mass disasters, even if each situation is peculiar (the damages reported and the resources available for resilience are different). In Turkey and Syria, there are geopolitical situations of conflict, the territory is narrow, and the number of displaced people has become uncontainable compared to other earthquakes in the world. 

The five pavilions approach is a framework for structuring emergency services personnel in an integrated disaster response, which can of course be shaped according to the needs of individual affected territories and, above all, in accordance with the recommendations imposed by local governments.

At any rate, it is crucial to have disaster site coordinators who know how to manage and bring together different professionals [29]. 

Looking to the future, finding systems that combine local knowledge with international experience to support governments in early-stage disaster responses is of great value. The United Nations Disaster Assessment and Coordination (UNDAC) model is a prime example of a multi-organisational operation [30]. The UNDAC system has predefined methods for establishing coordination structures and better organising information management during the first phase of a disaster. It is a proven system for mobilising and deploying a team that arrives at the disaster site within 12–48 h of request and is self-sufficient. The UNDAC strategy fits in with our proposal (immediate arrival, prompt assistance for recovery operations): the five proposed macro-areas of intervention cover all the possible operational strategies of a system such as that of UNDAC. In particular, our model would include an operational synergy between the various figures recruited on the disaster site (technicians, health workers, and volunteers). In these contexts, the ability to rapidly deploy multifunctional teams is essential. This deployment includes an extensive network of specialist partners with the range of expertise needed to support operational coordination and situational analysis through dedicated information management as well as specialist technical and logistical support. A systems approach can help all levels of government to organise disaster-related information in order to distinguish the important signals from the noise and improve decision-making across sectors, departments, and agencies [31]. 

The creation of teamwork between institutions or associations from different countries and collaboration in multi-professional teams is an important step in the organisation and coordination of health and care services in mass disasters [32]. Discovering new operational strategies, conducting empirical studies, generating new knowledge through practical exercises, and organising courses with expert staff can be very useful. Furthermore, the composition of different teams with interdisciplinary figures in the field of disaster research should be tested. Therefore, future studies could be designed to establish a series of scientific criteria or scores aimed at evaluating the efficiency and effectiveness of interventions among different professionals.

## Figures and Tables

**Figure 1 healthcare-11-02052-f001:**
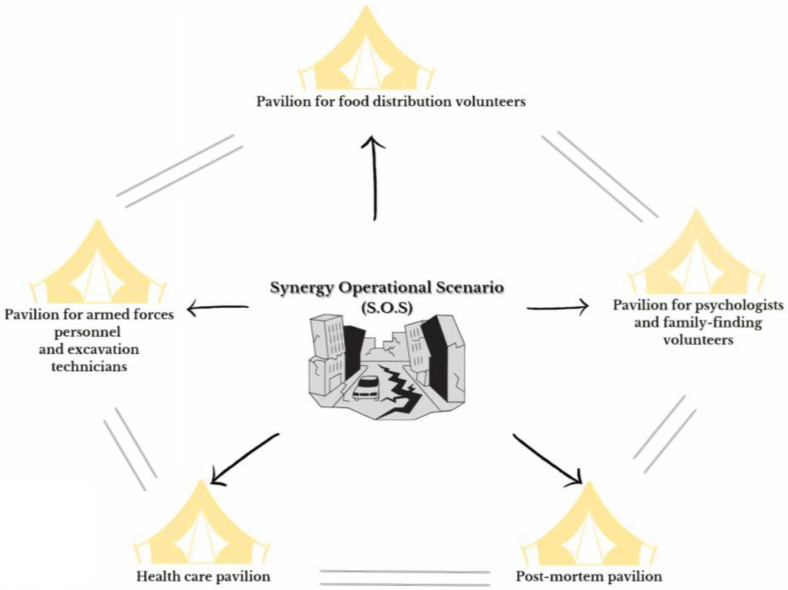
Model of operational health pavilions near the earthquake site.

## Data Availability

Not applicable.

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
