# Peer review of "Operational Health Pavilions in Mass Disasters: Lessons Learned from the 2023 Earthquake in Turkey and Syria"

_healthcare, 2023, doi:10.3390/healthcare11142052_

Round 1

Reviewer 1 Report

I want to thank the authors for considering "Healthcare" to publish their work. I added my comments here to better prepare the article for publication.

Comments are as below:

1-Line 46: I did not understand what do you mean by "based on the events that have occurred and the research that is still being done": what type of events and what type of research is still being done and how do you select organizational model structured around five pavilions?

2-Line 53 to 57: please be concise. You repeated one statement about three times emphasizing on the importance of time. Please restate and eliminate the duplications:

"Time is precious", "this is the most crucial phase of disaster management", "timely and effective deployment can limit damage and save lives", and "losses can be minimized and lives saved by fast and professional responders".

3-Line 57: please review the entire paper and remove unnecessary terms such as "Obviously". If something is obvious then there is no need to include it in your paper. If you are talking about something, then it is not obvious.

4-Please consider removing qualitative terms and include some facts from previous literature. 

Example: Lines 60 and 61: "There has been a marked improvement in coordination ....": Should I know what marked improvement is by myself? better to restate it like that: There is an increasing coordination between .... as is indicated in XX et al 20yy. 

Another example: line 62: "Much remains to be done to fill the gaps": if there is a specific gap in the literature, please pinpoint it. For example, unlike previous literature that focused only on the association between X and Y (references), this study aims to ......

5-I cannot see the match between the title and the paragraphs. For example, you have title 2: Pavilion for armed forces personnel and excavation technicians, and then in line 66 your paragraph talks about how body recovery should be done properly.  If you want to talk about both, please add another title and design subsections accordingly.

6-Line 81: again you started talking about the magnitude of the earthquake? You mentioned in line 27 that a powerful earthquake happened. Please follow an order and keep it consistent. My suggestion:

Line 27: talk about the scale and level of destruction, the significance of involvement, and the magnitude of the disaster. 

Line 81: Here, the readers are aware of the earthquake. You only want to talk about the healthcare workers. For instance: Given the magnitude of earthquake, all healthcare workers and the entire population of the area are involved.

Line 83 to 99: these are common knowledge. For example, without your paper, everybody knows that the hospital may or may not be located in the neighborhood of a disaster (line 83).

line 87: with the primary mission being to provide assistance: all of us know that the primary mission of EMSs is to provide assistance.

line 90: A pavilion needs to be set up quickly near the affected area: all of us know this. Is there anyone of your readers who thinks that setting up a pavilion should be delayed after a disastrous event??

line 98: provide high-quality support to people experiencing a health crisis. (Should it be low quality?)

General comment: You started with "introduction", and then I was expecting the "materials and method" and then "results and discussion" and or "implications". So I am confused about how you set your titles.

If you did a review paper: how you selected your articles, what is the time period? which database did you use? what is the method of screening and doing the qualitative analysis? (e.g., NVivo?)

Author Response

Dear Reviewer #1,

thank You very much for the review.

We are glad to inform You that all the criticisms have been accepted.

The manuscript has been revised and improved according to these criticisms.

In this regard, we really thank You for the relevant improvement to the article

This is our reply point-by-point.

  1. We made the events we referred to explicit by citing them in new references:

- Clark, M.A.; Hawley, D.A.; McClain, J.L.; Pless, J.E.; Marlin, D.C.; Standish, S.M. Investigation of the 1987 Indianapolis Airport Ramada Inn incident. J. Forensic Sci. 1994, 39(3), 644-649.

- Mohd Daud, S.M.S.; Mohd Yusof, M.Y.P.; Heo, C.C.; Khoo, L.S.; Chainchel Singh, M.K.; Mahmood, M.S.;Nawawi, H. Applica-tions of drone in disaster management: A scoping review. Sci. Justice. 2022, 62(1), 30-42.

- Forensic Anthropology and Medicine: Complementary Sciences From Recovery to Cause of Death / edited by Aurore Schmitt, Eugénia Cunha, João Pinheiro. - Totowa, NJ : Humana Press Inc, 2006.

- Marrone, M.; Tarantino, F.; Stellacci, A.; Baldassarra, S.L.; Cazzato, G.; Vinci, F.; Dell’Erba, A. Forensic Analysis and Identifi-cation Processes in Mass Disasters: Explosion of Gun Powder in the Fireworks Factory. Molecules 2022, 27, 244.

The organisational model was structured in 5 pavilions based on the Disaster Mortuary Operational Response Teams:

- U.S. Department of Health and Human Services. Disaster Mortuary Operational Response Teams. Available online: https://aspr.hhs.gov/NDMS/Pages/dmort.aspx#:~:text= Disaster%20Mortuary% 20Operational%20Response%20Teams%20(DMORTs)% 20support%20local%20mortuary% 20services,in%20a%20dignified%2C%20respectful%20manner

  1. We have redrafted the sentence: “In this stage, the timely, effective and efficient deployment of Operational Response Teams can pay off in terms of damage limitation and saving lives”.
  2. Correction done
  3. We have redrafted the sentence: “Undocumented removal of remains by volunteers, continuous and unsupervised access to the site and autopsy area, recovery of remains by family members prior to identification, non-involvement of forensic personnel and logistical criticalities occurred at the disaster site [10]”.
  4. The content of the paragraphs has been corrected to match the paragraph headings.
  5. The reference to the magnitude was removed having referred to it in the initial part of the introduction.
  6. Accepted suggestion: “Given the magnitude of earthquake, all health care workers and the entire popu-lation of the area are involved”.
  7. Deleted sentence
  8. Deleted sentence
  9. Deleted sentence
  10. Deleted sentence
  11. This manuscript is a brief report and not a systematic literature review. It started with the paragraph 'Introduction' in compliance with the MDPI guidelines.

Reviewer 2 Report

General Comments to Authors:

This piece is timely given the recent occurrence of the mass earthquake in Turkey and Syria. Though a Brief Report, the paper needs a Methods section as well as more perspective added into the Conclusion. A number of additional questions can be addressed that would make the piece even more relevant to an international audience. The language is overall well-written, though a few grammatical corrections do exist.

Methods section:

Address who developed the model, what are their qualifications, does the model have precedent in the literature or in the authors’ own work, was a literature search performed, were periodic meetings held to develop it, did points of disagreement or uncertainty exist in its development and how were they solved, were any previously existing segments of the model ultimately excluded?

Specific Comments to Authors:

P. 1, Para. 4, line 48:

Is this 5-part model a new one?

P. 2, Para. 2, line 54:

Mention readiness, suitable training of the military, resources – financial and otherwise – possessed by the military. Please add at least 1 line addressing need to coordinate efforts if military representation is multi-country. Could mention whether such was the case in the disaster being described.

P. 2, Para. 3, line 76:

Bodies [5].3. Results  ->  (This portion seems incomplete.)

P. 2, Para. 4, lines 77-79:

These lines are from Instructions to the Authors. They need to be deleted.

P. 3, Para. 1, line 105:

Major disasters in recent years have shown  ->  Major disasters in recent years, such as X and Y, have shown 

(fill in 1 or 2 examples)

P. 3, Para. 3, end of line 146:

Is the food resourced from the national level or multiple governmental levels?

P. 4, Para. 2, line 172:

Arranging funds  ->  Is this done ahead of time or in a manner coincident with the disaster?

P. 4, Para. 3, line 190:

line 190: of the associated health risks  ->  of the associated health risks of …   -or-   of the associated health risks (e.g., X, Y)

(supply an example or two of the associated health risks)

P. 5, Start of Para. 1:

The beginning of the second paragraph in the Conclusion (1st paragraph on p. 5) would benefit by addressing a number of questions: Regarding the call for an international task force, what kind of review is usually performed after such disasters? Is the request for such a task force different here, and if so, why?; Comment on what makes the Turkey / Syria situation unique.; Is the 5-part pavilion approach capable of further evolution in light of the task force recommendations that might accrue and further analysis of natural disasters?

P. 6, Alpert EA Reference:

2022 Jan -.  ->  (The end of this reference seems incomplete.)

Grammatical:

P. 2, Para. 3, line 68:

the use of procedures and health and safety precautions recommended in this manual  ->  the use of recommended health and safety precautions

(An academic article is not a manual; this wording needs to be dropped)

P. 2, Para. 5, lines 81, 86, 94:

healthcare  ->  health care

(“healthcare” refers to systems; “health care” to facilities and personnel)

P. 3, Para. 1:

line 114: e.g. body marks,  ->  e.g., body marks

line 115: This team will liaise with another  ->  This team often liaisons with another

lines 116-7: to contact the relatives and acquaintances of victims, in order to collect all the identity data  ->  to contact possible relatives and acquaintances of victims. This effort enables collection of all the identity data 

line 118: (start new paragraph with line beginning “It is useful …”)

line 120: activities, through  ->  activities through

line 122: (decolorize the yellow marking of “clothing.”)

line 125: should be filled in here, such as those  ->  should be filled in, such as those

P. 3, Para. 2:

line 137: , application  ->  , and application

line 139: and manner of death  ->  and determining the manner of death

P. 4, Para. 1:

line 149: healthcare  ->  health care

line 157: will be important  ->  is important

P. 4, Para. 2:

line 164: to keep calm and avoid panic attacks.  ->  to keep as calm as the situation permits and avoid panic.

line 167: stages will  ->  stages can

line 172: are particularly active  ->  have been particularly active

line 174: Local centers play an important role  ->  (The start of this line should begin a new paragraph.)

line 176: and by releasing  ->  and releasing

line 183: There are often  ->  (The start of this line should begin a new paragraph.)

line 186: organizations before  ->  organizations or divisions before

                will respond  ->  can respond

P. 4, Para. 3:

line 189: It is crucial to prepare the health sector  ->  As a vital component of the areas discussed, it is crucial to prepare the health sector 

line 194: decolorize “management.”

P. 5, Para. 1:

line 203: non-profit organizations of social utility on the front line  ->  non-profit organizations on the front line

(The reader already knows these organizations are socially beneficial; this intermediate phrase just lengthens the sentence.)

line 206: healthcare  ->  health care

line 207: etc.) it is  ->  etc.), it is 

line 201: Looking to the  ->  (The start of this line should begin a new paragraph.)

line 215: This includes  ->  This deployment includes

line 218: Systems approach  ->  A systems approach

See enclosed content / grammatical comments sheet.

Author Response

Dear Reviewer #2,

thank You very much for the review.

We are glad to inform You that all the criticisms have been accepted.

The manuscript has been revised and improved according to these criticisms.

In this regard, we really thank You for the relevant improvement to the article

This is our reply point-by-point.

1. The model was based on previous mass disasters that we have made explicit in the references:

- Clark, M.A.; Hawley, D.A.; McClain, J.L.; Pless, J.E.; Marlin, D.C.; Standish, S.M. Investigation of the 1987 Indianapolis Airport Ramada Inn incident. J. Forensic Sci. 1994, 39(3), 644-649.

- Mohd Daud, S.M.S.; Mohd Yusof, M.Y.P.; Heo, C.C.; Khoo, L.S.; Chainchel Singh, M.K.; Mahmood, M.S.;Nawawi, H. Applica-tions of drone in disaster management: A scoping review. Sci. Justice. 2022, 62(1), 30-42.

- Forensic Anthropology and Medicine: Complementary Sciences From Recovery to Cause of Death / edited by Aurore Schmitt, Eugénia Cunha, João Pinheiro. - Totowa, NJ : Humana Press Inc, 2006.

- Marrone, M.; Tarantino, F.; Stellacci, A.; Baldassarra, S.L.; Cazzato, G.; Vinci, F.; Dell’Erba, A. Forensic Analysis and Identifi-cation Processes in Mass Disasters: Explosion of Gun Powder in the Fireworks Factory. Molecules 2022, 27, 244.

It has its scientific basis in Disaster Mortuary Operational Response Teams:

- U.S. Department of Health and Human Services. Disaster Mortuary Operational Response Teams. Available online:https://aspr.hhs.gov/NDMS/Pages/dmort.aspx#:~:text=Disaster%20Mortuary% 20Operational% 20Response%20Teams%20(DMORTs)% 20 support%20local% 20mortuary % 20services, in%20a%20dignified%2C%20respectful%20manner (accessed on 20 April 2023)

2. Redrafted and implemented the sentence: “Readiness, adequate training of the armed forces, and the resources - financial and otherwise - possessed by the armed forces are required. It is also mandatory to coordinate efforts if the military representation is multinational”.

3. Word deleted (Instructions to the Authors)

4. Word deleted (Instructions to the Authors)

5. Bibliography included:

- Otero-Varela, L.; Cintora, A.M.; Espinosa, S.; Redondo, M.; Uzuriaga, M.; González, M.; García, M.; Naldrett, J.; Alonso, J.; Vazquez, T.; et al. Extended reality as a training method for medical first responders in mass casualty incidents: A protocol for a systematic review. PloS one, 2023, 18(3), e0282698.

7. Pavilion for food distribution volunteers: “The issue should always be addressed at both national and supranational level”.

8. However, it is necessary to plan ahead with regard to the treatment and management of cases of malnutrition that may have pre-existed the disaster, which may become manifest during the relief operation.

9. Redrafted and implemented the sentence: “There is an ongoing need to create awareness of the associated public health risks such as infectious diseases, and to enhance the knowledge and skills of all health actors involved in response and recovery processes”.

10. Redrafted and implemented the sentence: “After each disaster, a critical review of the activity should be carried out. This would provide an opportunity to learn from mistakes and improve forensic and medical work in mass disasters. In this perspective, the logistical-functional proposal formulated in this report is a proposal for improvement to standardise the multidisciplinary activity that characterises these scenarios”.

11. Corrected reference: Alpert, E.A.; Kohn, MD. EMS Mass Casualty Response. [Updated 2022 Aug 8]. In: StatPearls [Internet]. Treasure Island (FL): StatPearls Publishing; 2023 Jan-. Available from: https://www.ncbi.nlm.nih.gov/books/NBK536972/

12. All suggested grammatical corrections have been carried out. Many thanks to the Reviewer #2.

Round 2

Reviewer 1 Report

I reviewed both the initial submission and the revised version you provided. While I appreciate the effort you put into addressing our concerns, I believe that the paper still falls short in several key areas, making it unsuitable for publication in its current form. Allow me to provide you with specific feedback:

Body and Structure: The article lacks a clear and well-organized structure, which hinders the reader's understanding of the research. The introduction does not adequately set the stage for the study, and the subsequent sections lack coherence and fail to present a cohesive argument. We suggest reorganizing the paper to provide a logical progression of ideas and improve the flow of information. (your paper is more like a general newspaper which might not provide a contribution to the academic world).

Lack of Originality: The paper does not demonstrate sufficient originality in its contribution to the field. The research findings appear to replicate or restate existing knowledge without adding significant value. We encourage authors to conduct thorough literature reviews and highlight the unique aspects of their work to ensure its novelty and relevance.

Insufficient Information: The article lacks essential information necessary for readers to understand the research methodology and replicate the study. Key details, such as sample size, data collection methods, and statistical analyses, are either missing or insufficiently explained. Providing comprehensive and transparent information is crucial to ensure the rigor and reproducibility of the research.

Given these significant issues, I am unable to move forward with publication. However, I would encourage you to consider our feedback and revise your paper accordingly. Addressing the aforementioned concerns and seeking guidance from colleagues or professional editors may greatly enhance the chances of your work being accepted in the future.

Thank you for your understanding, and we wish you the best of luck with your future research endeavors.

Author Response

Dear Reviewer #1

Thanks for all your valuable suggestions. Our paper is a brief report and not an original article. However, in accordance with what was suggested, we modified the Introduction and, above all, we strengthened our contibution by inserting a Methods section and adding further references. To give originality, we have also added a figure (Fig. 1) to better represent our description. We have also implemented with further detail in the Conclusions section.

Our small manuscript arises from the assumption that in the event of a mass disaster, especially in less organized areas, immediate medical assistance is not always provided on disaster site. As said previously, it is a brief report which takes its cue from a recent disastrous earthquake that and, based on literature papers describing similar cases, proposes an organizational model built on operating pavilions which act in synergy to ensure immediate assistance and healthcare, starting with the disaster site.

Please check what we have added e reconsider our manuscript, that not requires a statystical analysis or a great number of references as an orginl article or a systematic review/metanalysis.

Thank yuo again for giving us this opportunity.

For details, please see the revised and updated file attached.

We hope that your comments have been met in this way.

We look forward to your reply.

Kind regards,

Dr. Francesco De Micco, M.D., Ph.D

Reviewer 2 Report

General Comments to Authors:

The paper has shown improvement, but is in need of a Methods section and further detail at the beginning of the Conclusions.

Specific Comments to Authors:

P. 2, After Para. 2, Needed Methods section (New section 2.):

Address, to whatever extent the authors are reasonably able, who developed the model, what are their qualifications, does the model have precedent in the literature or in the authors’ own work, was a literature search performed, were periodic meetings held to develop it, did points of disagreement or uncertainty exist in its development and how were they solved, were any previously existing segments of the model ultimately excluded?

P. 3, Para. 3, line 132:

Major disasters in recent years have shown  ->  Major disasters in recent years, such as X and Y, have shown 

(fill-in 1 or 2 examples)

P. 5, Start of Para. 2 in Conclusions:

The beginning of the second paragraph in the Conclusions (5th paragraph on p. 5) requires addition of further detail: Regarding disaster reviews, is the request for such a task force different here, and if so, why?; Comment on what makes the Turkey / Syria situation unique.; Is the 5-part pavilion approach capable of further evolution in light of the task force recommendations that might accrue and further analysis of natural disasters?

Grammatical:

P. 3, Para. 3, line 148:

activities, through  ->  activities through

P. 5, Para. 2, line 216:

Eliminate horizontal space between lines 216 and 217.

P. 7, Alpert EA Reference (#13):

2023 Jan -.  ->  2023 Jan. 

See enclosed comments.

Author Response

Dear Reviewer #2

all authors are grateful to you for considering our manuscript interesting and for your suggestions.

We accepted your proposals and implemented our manuscript.

  • We added a Methods section. Furthermore we have explained: “Our proposal arises both from what is reported in scientific literature and from one's own professional experience. In particular, we authors participated in inspections and post-mortem investigations on the occasion of mass disasters that occurred in Central Italy. We have also took part in national and international conferences on the topic.”. Finally, to give originality, we have also added a figure (Fig. 1) to better represent our description.
  • We have precised “Major disasters in recent years, such as 2010 Haiti earthquake or 2016 Central Italy earthquakes”.

  • The requested Task Force should not be different from those already activated in previous mass disasters (The United Nations Disaster Assessment and Coordination (UNDAC) model is a prime example of multi-organizational operation), even if each situation is peculiar (the damages reported and the resources available for resilience are different). In Turkey and Syria there are geo-political situations of conflict, the territory is narrow and the number of displaced people has become uncontainable, compared to other earthquakes in the world.

    The five pavilions approach is a framework for structuring emergency services personnel in an integrated disaster response, which can of course be shaped according to the needs of individual affected territories and, above all, in accordance with the recommendations imposed by the local Governments.

    At any rate, it is crucial to have disaster site coordinators who know how to manage and bring together different professionals.

    We have precised all these aspcets in the Conclusion section.

  • Finally we have corrected the grammatical tips.

  •  

For details, please see the revised and updated file attached.

We hope that your comments have been met in this way.

We look forward to your reply.

Kind regards,

Dr. Francesco De Micco, M.D., Ph.D
